# Systematic Review and Meta-Analysis of Electromyography Potential to Discriminate Muscular or Articular Temporomandibular Disorders and Healthy Patients

**DOI:** 10.3390/healthcare13050466

**Published:** 2025-02-21

**Authors:** Maria Isabella Federici, Francesca Di Pasquale, Chiara Valenti, Alessandro Nanussi, Ornella Tulli, Maddalena Coniglio, Stefano Eramo, Lorella Marinucci, Stefano Pagano

**Affiliations:** 1Faculty of Dentistry, Department of Medicine and Surgery, University of Perugia, S. Andrea delle Fratte, 06156 Perugia, Italy; mariaisabellafederici@gmail.com (M.I.F.); francesca.dipasquale@outlook.it (F.D.P.); ornellaefilippo@icloud.com (O.T.); conigliomaddalena@gmail.com (M.C.); stefano.eramo@unipg.it (S.E.); stefano.pagano@unipg.it (S.P.); 2CISAS, Center of Studies and Activities for Space, “Giuseppe Colombo”, University of Padua, Via Venezia, 15, 35131 Padua, Italy; 3Department of Interdisciplinary Medicine, University of Milano-Bicocca, 20126 Milan, Italy; alessandro@nanussi.it; 4Section of Biosciences and Medical Embryology, Department of Medicine and Surgery, University of Perugia, S. Andrea delle Fratte, 06156 Perugia, Italy; lorella.marinucci@unipg.it

**Keywords:** electromyography, stomatognathic diseases, musculoskeletal diseases, joint diseases, tem-poromandibular joint disorders

## Abstract

**Background/Objectives**: New devices such as surface electromyography (sEMG) have been proposed to support traditional gnathological examination and diagnostic protocols. The aim of this study is to investigate whether sEMG can be considered a diagnostic instrument to discriminate between healthy subjects and patients with temporomandibular disorders (TMDs) of an articular or muscular nature. **Methods**: A systematic review was conducted according to PRISMA guidelines using literature searches of MEDLINE (via PubMed), Scopus, and Web of Science. Inclusion criteria: recent clinical studies (≤10 years) in English or Italian, involving electromyography in TMD diagnosis, with a control group of healthy patients. Data considered to be homogenous were subjected to meta-analysis (95% confidence interval [CI]; α = 0.05). Hedge g was calculated because all variables were continuous. Articles meeting the inclusion criteria were checked for further consideration, and relevant data were collected into two tables. In total, 18 studies were included after full-text reading. Meta-analyses were carried out for the static impact index (IMP), percentage overlapping coefficient (POC), and torque coefficient, and dynamic Symmetrical Mastication Index (SMI). **Results**: Patients with TMD had lower values in all parameters except IMP. sEMG registered a reduction in masseter activity, lower chewing efficiency, and an increase in fatigue during contractions in TMD patients. **Conclusions**: sEMG is not reliable to distinguish healthy from TMD patients, but despite the limitations related to the high variability in the studies (type of electromyography, static or dynamic tests, and population characteristics), the sEMG results are reliable considering the POC and SMI parameters, encouraging more in-depth studies for a predictable clinical practice. Patients with TMD had lower values in the dynamic index SMI and in static indexes POC and torque coefficient, except IMP. EMG might performs better if employed in muscle forms.

## 1. Introduction

Temporomandibular disorders (TMD) comprise a number of various diseases which involve masticatory muscles, temporomandibular joints (TMJs), and/or their associated structures [1]. Epidemiological studies in several countries have shown a consistent incidence of orofacial pain symptoms in adults, demonstrating that approximately 40% of the population experience at least one of the symptoms of TMD [1,2], but etiology and pathophysiology are multifactorial and still unclear [3,4,5].

TMDs diagnosis is based on patient-reported symptoms, clinical consultation, and imaging, and the gold standard is the DC/TMD (Diagnostic Criteria for Temporomandibular Disorders), which is a validated protocol [6] based on the biopsychosocial model of pain, which consists of an Axis I physical evaluation using robust diagnostic criteria and an Axis II examination of psychosocial condition and pain-related impairment [6,7].

TMDs are described by symptoms such as muscular tensions, TMJ clicks, reduced mandibular movements, headaches, otalgia, and parafunctional habits [8], and can be classified in intra-articular or extra-articular. Musculoskeletal states are the most prevalent reason for TMDs, representing at least 50% of cases [9,10], while joint disc displacement affecting condylar–disc interactions is the most frequent intra-articular factor [11].

New devices have been proposed to support traditional gnathology examinations and established diagnostic protocols, overcoming operator-dependent error.

In particular, ultrasound, a non-invasive and cost-effective tool, has a clinically acceptable ability to diagnose disc displacement and joint effusions in patients with TMD [12] and is also used to guide the operator in botulinum masseter injections with the aim of reducing muscular TMD symptoms, managing oromandibular dystonia, myofascial pain syndrome, and tension-type headaches. Ultrasound allows for visualizations of the precise injection site, ensuring the accurate delivery of botulinum to the masseter muscle, avoiding the surrounding structures and minimizing adverse effects, and it is also a quantitative measurement tool able to compare the vertical height of the masseter muscle before and after botulinum injection [13].

Meanwhile, electromyography (EMG) has been proposed as a useful device to determine muscular functions during standardized tasks. EMG provides quantitative information on the muscles’ myoelectric output and records the electrical activity when the muscle fibres of a motor unit are activated during voluntary or involuntary actions.

EMG has been widely utilized across various medical fields, including neurology, psychology, psychiatry, physiatry, physical medicine, rehabilitation, kinesiology, and gnathology. Each discipline leverages specific aspects of this technique based on its clinical needs, and the degree in clinical acceptance of EMG varies depending on its application.

In kinesiology, sEMG is a well-established tool for analyzing movement disorders and is particularly useful in differentiating various types of tremors, myoclonus, and dystonia, as well as in evaluating gait and posture. In neurology and rehabilitation, sEMG is most commonly employed in the treatment of weak or paretic muscles due to peripheral nerve injuries [14]. EMG is also used to identify different pathologies, like amyotrophic lateral sclerosis or myasthenia gravis due to the degeneration of muscle tissue and nerve, for the assessment and neurorehabilitation of cerebral palsy in children [15]. It is used as part of physical therapy to enhance muscle activity and strength, both in post-operative rehabilitation following surgical nerve repair and in non-surgical cases [14], to evaluate muscle fatigue measurements in pre- and post-surgery monitoring [16], or in orthopedical applications, surgical procedures, nervous system studies, and postural assessments [17].

Meanwhile, in rheumatology, electromyography (EMG) is a valuable tool for supporting the diagnosis of inflammatory myopathy or neuropathic processes. It can help differentiate active inflammatory myopathy from glucocorticoid-induced myopathy, as virtually all patients with active myositis present with abnormal EMG findings. However, while EMG can confirm the presence of myopathy, it does not provide information on its etiology [18]. Regarding EMG findings in patients with osteoarthritis and rheumatoid arthritis, specific parameters, such as maximum molar bite force and maximum mouth opening range, may differ from those observed in healthy individuals [19].

Two approaches may be identified based on the receiving sensor type: intramuscular EMG and superficial EMG (sEMG). sEMG is a non-invasive instrument that can assess masticatory system variation using electrodes on the skin that process masticatory muscles bioelectric signals and record both static actions, such as isometric, resting, and clenching of teeth, and dynamic rhythmic actions, like chewing (Figure 1). Due to improved patient compliance, sEMG has been widely used in gnathology to analyze muscular conditions and identify alterations (Figure 2). However, evaluations of the reliability of muscle amplitude recording signals have been repeatedly questioned in scientific studies [20]. Biological variations, lack of repeatability of skin electrode placement, and recording artefacts may be confounding elements and explain conflicting results between different studies [21,22]. sEMG’s use as an exclusive diagnostic tool, and not only as a support for traditional techniques, is still controversial.

The amplitude of the mandibular muscle signal appears to be reduced in patients with TMDs [24]. While a recent study highlighted how subjects with myalgia have a higher average masseter electromyographic amplitude than patients with painful TMDs, probably due to the high frequency of oral parafunctions [25], another study revealed instead that a raw resting sEMG assessment is able to distinguish between healthy and TMD subjects with moderate sensibility [26]. However, a review concludes that there are no supporting data for the application of sEMG to diagnose TMDs [20]. It would be necessary to refer to more recent review papers; moreover, studies examining EMG use to discriminate articular TMD from muscular TMD are also lacking in the literature.

The aim of this systematic review and meta-analysis is to understand whether sEMG can be considered a valid diagnostic instrument to distinguish both healthy patients from TMD-affected patients and, in particular, patients with articular TMD from those with muscular TMD.

## 2. Materials and Methods

This review was conducted considering the preferred reporting items for systematic review and meta-analysis (PRISMA 2020 statement) (Appendix A) [27], and a PRISMA flow diagram was used to report the study inclusion process (Figure 1). The protocol is available upon contacting the authors, and it has been registered online at the Open Science Framework (OSF) registries “https://osf.io/t67jd (accessed on 21 May 2024)”.

This review was performed following the Population, Inclusion, Comparison, Out-come, Study design (PICOS) format, according to the following question: “What is the ability of sEMG to discriminate between healthy or TMD patients, of muscular or joint nature?” [28].

Inclusion criteria: recent clinical studies (≤10 years), in English or Italian, involving electromyography in TMD diagnosis, with a control group of healthy patients.

Exclusion criteria: studies without controls, case reports, conferences, commentaries, editorials, guidelines, reviews and meta-analyses, discussions and opinions, animal studies, studies with patients in psychiatric therapies, previous traumas, ongoing orthodontic or TMDs treatments, systemic pathologies, and patients with a history of maxilla-facial surgery treatment. Studies that did not have a full text were also eliminated [27].

On the basis of the MeSH terms from PubMed, adapted to elaborate the search strategy, databases searches were performed in MEDLINE (via PubMed), Scopus, and Web of Science (Appendix A). After eliminating duplicates and non-relevant articles, the titles were independently selected by two reviewers according to the eligibility criteria.

Articles that met the inclusion criteria were first screened based on their abstracts. The full texts of potentially eligible articles were then reviewed independently by four reviewers. Data extraction was also performed independently by the same reviewers, and any discrepancies or uncertainties were resolved through discussion with the other authors to reach a consensus. Final decisions were made through group discussions, ensuring consistency and accuracy in the extracted data. Following the electronic search, the reviewers conducted a manual search of the reference lists of the included articles, applying the same inclusion/exclusion criteria.

Relevant data were organized into two tables, including the following:

Table 1: author(s), year of publication, EMG parameters evaluated, and types of EMG tests analyzed;

Table 2: country, number of cases and controls, type of electromyography device, diagnostic criteria, inclusion and exclusion criteria, main results, conclusions, and funding sources.

All variables were continuous and described by mean ± standard deviation (SD). The statistical software program Stata 18 (StataCorp LLC, Texas, TX, USA) was used. A 95% confidence interval (CI) was selected to compare EMG parameters’ values between non-TMD and TMD patients (α = 0.05). Hedge’s g measure was estimated considering the difference between the experimental group and the control group in terms of SD. Forest plots were used to graphically visualize the results. The I^2^ statistic was taken into account to assess the heterogeneity of the included studies: with I^2^ < 25% the rating was low, with 25% < I^2^ < 75% it was moderate, and with I^2^ > 75% it was high. The statistical analysis was performed with a random-effects model for I^2^ > 0% and with a fixed-effects model for I^2^ = 0%.

Three independent reviewers carried out the quality assessment of the included studies using a Quality Assessment Tool with diverse studies (QuADS) [29]. With this evaluation tool, the methodological quality of the included studies and the extent to which a study addressed the possibility of bias in its design were evaluated. The QuADS tool considered details on the rationale and aim, subjects and setting, study design, sampling and recruitment, data collection, exposure measurements, analysis methods selected, stakeholder involvements, and limitations. These 13 evaluation criteria were rated on a scale from 0 to 3 (0: absence of the element; 1: very limited presence of the element; 2: moderate presence of the element; 3: complete and adequate presence of the element) (Appendix A).

## 3. Results

### 3.1. Characteristics of the Included Studies: Country, Year of Publication, and Sample Size

A total of 421 potentially eligible papers were found after the electronic strategy search. In total, 190 articles were eliminated after duplicates removal, and 231 studies were considered from the title/abstract reading: 177 studies were excluded, and 54 studies were included for full-text review. Finally, 18 studies were included (Figure 3).

Three articles were published in 2014 [26,30,31], three in 2015 [32,33,34], three in 2016 [35,36,37], four in 2017 [38,39,40,41], one in 2018 [42], two in 2020 [43,44], one in 2021 [45], and one in 2022 [46]. Ten studies had a setting in Brazil [30,32,33,35,36,37,39,40,42,46], two in China [38,43], three in Italy [31,44,45], one in Spain [26], and two in the USA [34,41] (Table 2).

Considering the participants involved in the studies, two articles evaluated children [39,40], while all the other studies investigated an adult population; eight studies included only females [32,36,37,41,42,44,45,46], whereas all other studies evaluated participants of both sexes. All the included studies evaluated masseter and anterior temporalis muscle activity, while suprahyoid muscle activity was investigated in only two studies [40,42].

### 3.2. Quality Assessment Score

All of the 18 articles included met the criteria of the quality assessment, resulting in being reliable with a low risk of bias (Appendix A). The highest score was 35/36 [39,44], and the lowest was 25/36 [34].

### 3.3. EMG Parameters Analyzed

After static and dynamic tests, the following parameters were analyzed: intraindividual indexes of sEMG activity between masseter and temporalis muscles (activity index) [26,31,45], between both side muscles (SIDE) [26], torque coefficient [26,31,44,45], the proportionality of the values for every muscle and their average (or percentage index) of the sEMG between clenching and resting [26], asymmetry index (ASIM) [26,31,44], muscular centre of gravity (BAR), total standardized muscle activity or impact index (IMP), percentage overlapping coefficient (POC), and Symmetrical Mastication Index (SMI) [44], and the area under the ROC (AUC) [31] (Table 1).

**Table 1 healthcare-13-00466-t001:** Type of tests and parameters evaluated in the included studies.

	Tests							Parameters									
	Static				Dynamic												
	MVC Isometric	MVC on Parafilm	MVC on Cotton Rolls	MVC on Intercuspal Position (Clench)	Duty Factors	Chewing	Resting and Clenching Tasks	Activity Index	Torque	Percent Index	ASIM	SIDE	BAR	IMP	POC	SMI	Area Under Roc Curve (AUC)
Berni, 2015 [32]		x															x
Chaves, 2017 [40]		x															
De Paiva, 2022 [46]	x																
Di Giacomo, 2020 [44]			x	x					x		x		x	x	x	x	
Ferreira, 2014 [30]			x											x		x	
Hu, 2020 [43]				x													
Iwasaki, 2015 [34]					x												
Iwasaki, 2017 [41]					x												
Lodetti, 2014 [31]			x	x				x	x		x (evaluated using POC)						
Mapelli, 2016 [35]			x												x		
Pires, 2018 [42]		x															
Politti, 2016 [36]	x	x															
Ries, 2016 [37]		x															
Rodrigues, 2015 [33]						x											
Santana-Mora, 2014 [26]							x	x	x	x	x	x					
Serrano, 2017 [39]		x															
Valentino, 2021 [45]			x	x				x	x					x	x		
Xu, 2017 [38]			x (one side)														

### 3.4. Principal Findings

Indices are highly encouraged to be used to improve the discrimination capacity of the sEMG analysis, and only one study indicated a moderate sensibility to differentiate between healthy subjects and TMD [26].

Berni et al. and Rodrigues et al. stated that TMD patients have a significantly higher activity in muscles than controls [32,33], while Mapelli et al. instead showed a significantly lower maximal activity in TMD patients for temporalis and masseters [35].

Not only patients with myogenous TMD have a reduced masseter electrical task [42], but also subjects with disc displacement (DD) generally used their temporals and masseters muscles at low levels compared to patients with DD and pain or patients with only pain; in particular, subjects with DD and pain used significantly higher temporalis activity compared to masseter muscles [34]. Patients with bilateral DD also have greater TMJ energy density, masseter, and temporal muscle duty factors and mechano-behavioural scores [41].

Considering children with TMD, a lower EMG activity than controls is highlighted, with lower bite force and significant difference for masseter and temporal muscles in MCV [39]. During MVC on parafilm, TMD patients reported significantly reduced masseters activity and higher suprahyoid activity [32] and, during clenching, an increased anterior temporalis activity and reduced masseters activity compared to an asymptomatic patient, as a consequence of nociceptive inputs [40]. Hu et al. also highlighted a significantly greater mean clenching level in healthy subjects [43].

Ries et al. reported a progressive increase in susceptibility to bilateral fatigue of superficial masseters and anterior temporalis during contraction for TMD patients compared to controls, with a considerable increase after 5 s and 10 s of maximal contraction [37]. In addition, significantly higher median power frequency values were highlighted for right and left masseter and temporalis [36] and for suprahyoid muscles [42] on TMD patients than healthy subjects.

Considering asymmetry, Di Giacomo et al. showed how there was no differentiation from the TMD group to controls [44], but other studies revealed how TMD subjects exhibited left-dominant asymmetry compared to healthy patients [46], particularly on temporalis muscles [35], and an asymmetrical activation of jaw-closing muscles [45]. TMD patients also have a significantly lower coordination between masseter and temporalis muscles [35] and lower chewing efficiency than controls [44], but stronger jaw-closing muscles than controls [45].

### 3.5. Electromyographic Indices Included in the Meta-Analysis

Torque, IMP, POC, and SMI were selected for the meta-analysis. In general, the index I^2^ was used as a reference because it does not depend on the number of studies contained in the meta-analysis and therefore it turns out to be more reliable. However, in meta-analyses with a very small number of studies, the effect of heterogeneity could be overestimated.

Regarding torque, TMD patients showed lower values than the control ones, and the heterogeneity indices showed perfect homogeneity across studies. The effect size was not statistically significant (Hedge’s g with 95% CI = −0.13 [−0.40, 0.15]) (Figure 4). The funnel plot showed no problems with publication bias (Appendix A).

Taking into consideration IMP, TMD patients had greater values than the control ones, the two studies were highly heterogeneous, the effect size was not statistically significant (Hedge’s g with 95% CI = 0.16 [−0.17, 0.49]) (Figure 5), and it did not make any sense to consider publication bias in this situation (Appendix A). The IMP meta-analysis obviously has the major limit on the number of studies included, and therefore the value of I^2^ = 76.54% has to be considered an overestimate in respect to the real heterogeneity.

As far as POC is concerned, Di Giacomo reported no substantial difference [44], while Lodetti et al. pointed out how the coefficient was quite high for temporalis and masseter, showing a significant discrimination capacity between osteoarthritis and soft tissue injury [31]. TMD patients showed lower values than healthy ones, there was perfect homogeneity between studies, the effect size was statistically significant (Hedge’s g with 95% CI = −0.22 [−0.39, −0.06]) (Figure 6), and the funnel plot did not show publication bias issues (Appendix A).

TMD patients instead had lower SMI values than controls, and there was some heterogeneity among studies, but this did not seem pathological since the results of the studies appear globally consistent with each other. The effect size was statistically significant (Hedge’s g with 95% CI = −0.72 [−1.06, −0.38]) (Figure 7), and the funnel plot did not show publication bias issues (Appendix A). SMI analysis has an I^2^ equal to 35.25%; this value highlighted the presence of a certain heterogeneity. It could be hypothesized that there is a problem regarding the difference in the sample size between Mapelli et al. [35] and the other studies included in the meta-analysis [30,44].

The fixed-effect model was chosen for IMP meta-analysis (Figure 5) including only two studies. For all other meta-analyses (Figure 4, Figure 6 and Figure 7), it was considered more appropriate to use the random effects model, as the included studies present methodological and contextual differences that lead to excluding the possibility of a fixed-effect model.

Regarding the funnel plots (Appendix A), the same analysis was applied to all study groups, and none of the plots suggest significant publication bias. The only exception is the POC analysis (Appendix A), where one study falls outside the confidence interval. However, this was not considered a substantial issue, and, therefore, additional analyses such as Egger’s test were not performed.

**Table 2 healthcare-13-00466-t002:** Characteristics of the included studies.

Author	Country	Cases (TMDs)	Controls (Healthy TMJ)	Electromyography Devices	Diagnostic Criteria	Inclusion Criteria	Exclusion Criteria	Main Results and Conclusions	Fundings
Berni, 2015 [32]	Brazil	80 women	43 women	EMG1000, signal acquisition module (Lynx Tecnologia Eletrônica Ltd.a, São Paulo, SP, Brazil).	Myofascial pain (Ia), myofascial pain with limited mouth opening (Ib), IIa: disc displacement with reduction; IIc: disc displacement without reduction and without limited mouth opening; IIIa: arthralgia. Clinical exam according to Axis I of the RDC/TMD.	Full natural permanent dentition (28 teeth at least).	Tooth loss, BMI > 25kg/m^2^, systemic disease (arthritis, arthrosis, or diabetes), history of trauma to the face or TMJ, history of subluxation or luxation of the TMJ, current orthodontic treatment, or current medical treatment involving an anti-inflammatory agent, analgesic, or muscle relaxant.	TMD patient exhibited significantly greater muscular activity compared to the controls. In particular, during MVC on parafilm, the TMD group exhibited significantly lower activity in the masseter muscles and significantly greater activity in the suprahyoid muscles.	None
Chaves, 2017 [40]	Brazil	17	17	Myosystem^®^ Br-1. (Data Hominis Ltd., Uberlandia, MG, Brazil)	Axis I RDC/TMD.	Children of both sexes between 7 and 12 years of age.	Using any orthodontic appliances, mouth breathing history, had undergone orthodontic treatment, had systemic or rheumatologic diseases, or had a history of cervical spine or temporomandibular joint (TMJ) trauma or malformations. In the CG, children with a history of orofacial pain (related to TMD or not).	Lower sEMG ratio between masseter (M) and anterior temporalis (TA) muscles during clenching with increased activity of TA and reduced activity of M muscles in children with TMD compared to asymptomatic ones. Children with TMD preferentially used TA muscles during maximum voluntary clenching to obtain pain relief.	None
De Paiva, 2022 [46]	Brazil	72	30	Surface Meditrace™ (Ludlow Technical Products, Gananoque, Canada) Ag/AgCl electrodes and MiotecSuite^®^ software (Miotec Biomedical Equipments, Porto Alegre, Brasil).	Diagnosis of TMD arthralgia associated with myofascial pain, according to the RDC/TMD axis I.	Minimum of 28 permanent teeth and age between 18 and 45 years.	Trauma to face and TMJ, systemic diseases such as arthritis, fibromyalgia, pain due to migraine, headache or neck pain unrelated to TMD, ongoing use of analgesics, anti-inflammatory drugs, muscle relaxants, or psychotropic medications, acute infections or other serious dental, ear, eye, nose, or throat conditions, and neurologic or cognitive deficits.	TMD patients exhibit increased TA muscle ratio (*p* = 0.007) and an asymmetry of left dominance, compared with healthy subjects for both the TA (*p* = 0.02) and M muscles (*p* = 0.001).	Funded by Fundação de Amparo à Pesquisa de Minas Gerais–FAPEMIG, grant number APQ 02040/18, by the Coordenação de Aperfeiçoamento de Pessoal de Nível Superior—Brazil (CAPES)—Finance Code 001, and the Federal University of Juiz de Fora, supporting the APC
Di Giacomo, 2020 [44]	Italy	64 women	40 women	Wireless EMG device (TMJOINT, BTS SpA, Garbagnate Milanese, Italy).	DC/TMD (based on the biopsicological model of pain) Axis I.	Volunteer women students with unilateral TMJ/Disc Displacement with Reduction (DDR).	Other types of joint disorders (locking), Instable medical or psychiatric illness, positive history for a substance abuse in medical anamnesis, neurological disorders, craniofacial syndromes, history of local or general trauma, pregnancy, absence of teeth, fixed or removable prothesis, and current orthodontic or dental treatment.	No substantial difference between TMD patients and controls with muscles symmetry (POC and ASIM). No significant difference between the average IMP values. TMJ/DDR had lower chewing efficiency compared to controls (alteration of BAR index).	None
Ferreira, 2014 [30]	Brazil	46 women	30 women	Disposable silver/silver chloride bipolar surface electrodes (diameter 10 mm, interelectrode distance 21 ± 1 mm; Double; Hal Ind. Co., Ltd.a., São Paulo, SP, Brasil).	TMD, according to the Research Diagnostic Criteria for TMD (RDC/TMD), axis I, with chronic and moderate–severe symptomatology based on the ProTMDmultipart II questionnaire. Bilateral TMD, with muscle diagnoses (RDC/TMD group I) associated with disc displacement with reduction (DDR). (RDC/TMD group, or DDR and arthralgia, or arthralgia alone.	TMD group: To present TMD, according to the Research Diagnostic Criteria for TMD (RDC/TMD), axis I with chronic and moderate-severe symptomatology. Control group: To present Angle occlusal Class I, overbite and overjet between 2 and 4 mm, and no TMD based on the RDC/TMD.	Tooth absence, denture use, crossbite, dental pain or periodontal problems, pregnancy, neurological or cognitive deficits, previous or current tumours or traumas in the head and neck region, current or previous orthodontic, orofacial myofunctional, or TMD treatments, and current use of analgesic, anti-inflammatory, and psychiatric drugs.	SMI was lower in TMD patients (*p* < 0.05).	None
Hu, 2020 [43]	China	42	36	BioPAK EMG system version 7.2 with optically differential amplifiers.	RDC/TDM diagnoses in group II.	All volunteers for sEMG examination (informed consent) who had disc displacement were included in this study.	Occlusion was not included as a factor because it was beyond the scope of the study to conduct such an evaluation.	No significantly different mean resting sEMG levels between healthy subjects and patients (*p* > 0.05). Mean clenching sEMG levels significantly different (*p* = 0.003 < 0.05) than the mean for healthy subjects greater than TMD patients. The range in mean values of sEMG (M and TA) during mandibular rest were larger in healthy subjects. Higher levels of sEMG activity during rest were caused by artefact activity.	None
Iwasaki, 2017 [41]	Missouri (USA)	29 women with bilateral TMJ disc displacement (DD);	18 women with no DD	sEMG custom portable with pre-gelled disposable surface EMG electrodes (Alpine Biomed, Tonsbakken, Denmark).	Diagnostic Criteria for TMD (DC/TMD) with magnetic resonance (MR) and computed tomography (CT) to evaluate bilateral TMJ DD.	Bilateral TMJ disc displacement (DD).	Unilateral TMJ disc displacement and history of trauma and evidence of degenerative hard tissue changes in the TMJ.	TMJ energy density (*p* = 0.012), M and TA muscle duty factors (*p* < 0.01), and mechano-behavioural scores (*p* < 0.04) were significantly larger in women with bilateral TMJ DD compared to healthy subjects and support a general fatigue model for TMJ DD.	National Institute of Dental and Craniofacial Research (R01 2DE016417, JN-PI)
Iwasaki, 2015 [34]	USA	32	39	sEMG portable (DISA elektronik Denmark, Copenhagen, Denmark).	RDC/TMJ.	14 patients bilateral disc displacement without TMD pain; 18 patients with bilateral disc displacement and chronic myofascial and or TMJ pain.	Pregnancy, systemic muscoloskeletal or rheumatological disease such as fibromyalgia or muscular atrophy, TMJ-degenerative disease based on CBCT imaging, multiple missing teeth, large dental restoration, or fixed orthodontic appliances.	Women with disc placement (DD) used their TA and M muscles at low levels for a significantly greater percentage of time compared to women with DD and chronic pain (P) and women and men without DD. Significantly more night-time M muscle activities at low levels of jaw loading in +DD-P women compared to other groups. Patient with DD and pain used significantly higher TA compared to M muscle activity to produce 20N bite forces.	Supported by NIDCR
Lodetti, 2014 [31]	Italy	24	38	Computerized instruments (Freely, De Götzen srl, Legnano, Milano, Italy).	RDC/TMD Axis I (clinical and radiographic assessment); Axis II (to evaluate psychological status and pain related disability.	Patients with disc displacement (DD) and patients with osteoarthrosis and/or disc displacement (OA), a long lasting TMD (duration of symptoms longer than 6 months), pain in one or both joints (larger than 4 on 1–10 VAS) during mastication, limitation during opening (max non forced opening < 30 mm) or during left or right excursion or protrusion (<7 mm) at last molar maxillary-mandibular contact per dental hemiarch.	Presence of congenital craniofacial anomalies and or systemic disease, dental pain, periodontal problems, craniofacial and cervical trauma and surgery, mono o bilateral posterior edentulism, or current orthodontic treatment.	Concerning the association of age and z scores of the EMG indices with the MRI score, the correlation coefficient was quite high for POC TA (r = 0.85, with 95% C.I., 0.69–0.93) and moderately high for TC (r = 0.57, 95% C.I., 0.21–0.79) and POC M (r = 0.46, 95% C.I., 0.07–0.73), while for age and the other z scores, the coefficient was not significant at the level of 5%. Each of the z scores showed a significant and moderately high ability in discriminating between osteoarthrosis and damaged soft tissues, with AUROCs ranging from 0.73 to 0.79. Subject age showed a high discriminating ability with an AUROC of 0.91.	None
Mapelli, 2016 [35]	Brazil	30	15	Wireless EMG system FreeEMG, BTS S.p.A., (Garbagnate Milanese, Italy), with light probes (weight, 5 g) clipped to the electrodes.	RDC/TMD, axis I.	Chronic TMD pain > 6 months (myalgia and/or arthralgia) with DDR diagnosis, based on history and clinical examination; for the control group, criteria were good general health and the absence of TMD history.	Those with tooth absence (except the third molars), dental pain or periodontal problems, denture use, dentofacial deformities, crossbite, open bite, pacemaker use, neurological or cognitive deficits, previous or current tumours or traumas in the head and neck region, pregnancy, current or previous orthodontics, orofacial myofunctional or TMD treatments, current use of analgesic, anti-inflammatory, and psychiatric drugs were excluded from the study.	TMDs patients had a significantly lower maximal activity of the TA (*p* = 0.039) and M muscles (*p* = 0.025) and larger asymmetry on TA (*p* = 0.021) than controls. Also, both patient groups showed smaller coordination between the pairs of M and TA muscles compared to controls (POCTM: TMDmo, *p* = 0.038; TMDse, *p* = 0.008) and larger TA muscle prevalence (Asynergy index: TMDmo, *p* = 0.026; TMDse, *p* = 0.026). During chewing, significant inter-group differences were found for DMcenter, wActivity, and SMI. Compared to controls, the TMDse had significantly lower median working side M prevalence (lower differential working–balancing side activity of M muscle) (*p* = 0.042), that also induced a lower wActivity (*p* = 0.026) together with a worse degree of symmetry (SMI) between chewing performed on the right and left sides (*p* = 0.012).	None
Pires, 2018 [42]	Brazil	74 women with myogenous TMD (mean age: 26.54 ± 2.45 years)	30 asymptomatic women (mean age: 25.85 ± 2.57 years)	BIO-EMG 1000 electromyograph (Lynx^®^, Tecnologia Eletronica Ltd.a., São Paulo, Brazil).	Myogenous TMD [myofascial pain (Ia) or myofascial pain with limited mouth opening (Ib) according to the RDC/TMD.	Age between 18 and 40 years, diagnosis of myofascial pain with bilateral TMJ dysfunction, pain and/or fatigue in the masticatory muscles during functional activities for more than 6 months, and body mass index (BMI) between 18 and 25 kg/m^2^.	Tooth loss, BMI greater than 25 kg/m^2^, systemic disease (such as arthritis, arthrosis, or diabetes), trauma to the face or TMJ, sub-luxation or luxation of the TMJ, current orthodontic treatment or current medical treatment involving an anti-inflammatory agent, analgesic, or muscle relaxant, and volunteers with a diagnosis of arthrogenous TMD (IIIb and IIIc) according to the RDC/TMD.	Reduction in electrical activity in patients with myogenous TMD of M muscles and increased firing rate of the motor units of the suprahyoid muscles: Integrated EMG signal (IEMG) values significantly higher in the M than TA in the controls (*p* < 0.01). IEMG values significantly higher in the M of the controls than the group with myogenous TMD (*p* < 0.05); median power frequency values of the suprahyoid muscles were significantly higher in the myogenous TMD group than the controls.	None
Politti, 2016 [36]	Brazil	54	27	EMG System do Brazil LTDA (São Paulo, Brazil).	RDC/TMD Axis I and II.	Cases group: see diagnostic criteria. Control group: subjects without any stomatognatic function abnormality, full natural permanent dentition, absence of pain, and/or fatigue in the anterior temporalis and/or masseter muscles. Angle class I occlusion.	Missing teeth, open bite, overbite, crossbite, use of dentures, use of orthodontics, use of orthopedic appliances, history for trauma to the face, history of fracture in lower limbs or back, systemic condition (diabetes, arthritis), alteration in the vestibular system, systemic neuromuscolar disease, being currently in physical therapy, dental, or medicinal treatment, and ingestion of alcoholic 24 h prior to the evaluation.	Median frequency values of TMD patients were significantly higher (*p* < 0.05) than healthy subjects for all of the muscles assessed: both right and left M and TA.	This study is supported by Universidade Nove de Juhlo (UNINOVE Brazil) AND BRAZILIAN Fostering agencies Fundacao de Amparo a Pesquisa
Ries, 2016 [37]	Brazil	26	23	Miotool, MIOTEC, Porto Alegre, Brazil.	RDC/TMD.	Full permanent dentition with at least 28 teeth.	Those with a history of trauma to the face, temporomandibular joint, neck, and scapular region, vestibular alterations, dislocation, systemic disorders such as arthritis and arthrosis, use of orthodontic and/or orthopedic functional appliance, and the use of analgesics and anti-inflammatory drugs were excluded from the study.	Progressive increase in susceptibility to fatigue during the contraction for both superficial M and TA muscles bilaterally in both groups, with a significant increase after 5 s and 10 s of maximum contraction. A significant recovery was observed for both muscles in both groups during rest, especially after 5 s and 10 s. Compared to the controls, TMD subjects presented more levels of susceptibility to fatigue, with a significant difference in the right M and right TA muscles during the course of contraction.	None
Rodrigues, 2015 [33]	Brazil	27	25	Myotrace 400; Noraxon Inc., Scottsdale, AZ, USA	RDC/TMD—Axis I	Patients aged between 18 and 60 years old and without any treatment for TMD were included in the study.	Patients wearing dentures, who were toothless or had major dental absences, patients with trauma history in head and neck, with psychiatric and/ or neurological disorders, with rheumatic diseases, using drugs (antidepressants, muscle relaxants, anxiolytics, and anticonvulsants), and pregnant women were excluded.	Controls showed higher balance between activities of the muscles compared to the TMD group, which showed greater activity of TA muscles compared to M. All tested muscles showed greater activity in the TMD group compared to the controls, with a significant difference (*p* = 0.05).	None
Santana-Mora, 2014 [26]	Spain	53 (52 females and 1 male) patient with chronic unilateral TMD, from 15 to 55 years (mean age of 22.13 (7.27) years)	38 healthy students with no TMD (35 females and 3 males) with mean age 20.40 (1.32) years	Electrodes connected to Nicolet Viking Select electrodiagnostic system (Nicolet Biomedical, Middleton, WI, USA)	Unilateral Axis I TMD according to RDC/TMD	Fully dentate with normal occlusion and be right-handed.	Periodontal pathology, caries or damaged dental tissues, orthodontic therapy, maxillofacial disease, botulinum A toxin therapy, and psychological disorders.	Raw sEMG evaluation of rest provided moderate sensitivity and specificity to discriminate between healthy and TMD patients. The use of the indexes (mainly assessing the dominance of TA over masseter M during rest) is recommended to increase the discriminatory capacity of sEMG evaluation.	PI11/02507 from the Institute of Health Carlos III and MTM2011-28285-C02-00from the Ministry of Science & Innovation of the Government of Spain
Serrano, 2017 [39]	Brazil	45 children (mean age 8.8 years; 22 boys and 23 girls), divided in 3 groups: GII with mild TMD (18), GIII with moderate TMD (12), and GIV with severe TMD (5)	10 children	Electromyographer Myosystem-Br1 (DataHomins Ltd.a, Uberlandia, MG, Brazil.)	Faces Pain Scale-Revised (FPS-R) to determine the level of severity of the signs and symptoms of TMD Axis I of the RDC/TMD	Children with signs and symptoms of TMD without preference for gender or ethnic group.	Systemic diseases, ongoing treatment with medications that affect muscular activity (antihistaminic, anxiolytic, homeopathic, or other drugs with suppressive action on the central nervous system), uncooperative behaviour, history of trauma, dental pain, orthodontic treatment, otorhinolaryngolog-ical treatment, or speech therapy.	Children with TMD signs and symptoms had lower EMG activity than controls; significant difference among the groups for the left TA at rest (*p* = 0.01), right (*p* = 0.03) and left (*p* = 0.05) laterality, and for the left M (*p* = 0.01) and left TA (*p* = 0.03) muscles in maximum voluntary contraction. The bite force was lower in the TMD groups than controls, with a significant difference for the right region (*p* = 0.03).	National Council for Scientific and Technological Development, Brazil (CNPq-process n. 134281/2007-1)
Valentino, 2021 [45]	Italy	45	18	EMG system with wifi probes clipped to the electrodes TMJOINT (BTS Spa, Garbagnate Milanese, Milano, Italy)	RDC/TMD	Diagnosis of TMD myalgia with self-reported pain from at least 6 months.	(1) Neurological disorders. (2) Craniofacial syndromes. (3) Current orthodontic or dental treatment. (4) Pain in the jaw from 30 days.	Both this group and the control have asymmetric activation in the jaw closing muscles. Patients with myalgia had greater muscular work than controls. Difference between group and control for ATTIV index. IMPACT was greater in myalgia patients. TMD patients had harder jaw-closing muscles than controls.	None
Xu, 2017 [38]	China	15	13	Noraxon 1400a (USA)	The inclusion criteria for the TMD group were as follows: diagnosis of a muscle-related TMD of type Ia (myofascial pain) or Ib (myofascial pain with limited opening of the mouth), according to the Research Diagnostic Criteria for Temporomandibular Disorders Axis I, unilateral TMD.	No current TMD treatment, no current orthodontic treatment, mouth can be opened beyond 20 mm, pain duration was >6 months, and no missing teeth.	Neurological or cognitive deficit, dental pain, or periodontal problems.	Pre- and post-maximal bite forces were lower in TMD patients than controls (*p* < 0.01). The two-way factorial ANOVA indicated a significant effect (*p* < 0.05) of the group (for the M and TA muscles during the beginning, M muscles during the middle, and TA muscles during the end of the fatigue test) and side (only for TA muscles during the middle of the fatigue test). The TMD patients showed higher normalized RMS than the healthy subjects (*p* < 0·05). In TMD patients, ANOVA showed a significant decrease in median frequency for the clenching side of the M (*p* < 0.01) and TA muscles (*p* < 0.01) and balancing side of the M muscle (*p* < 0.01) during the fatigue test.	N/A

## 4. Discussion

TMDs are functional disorders of the chewing apparatus [47]. As there are similarities in symptoms between the TMDs and other conditions that cause orofacial pain, researchers tried to make accurate differential diagnosis, improving the diagnostic tools in addition to the clinical examination, the diagnostic search criteria for temporomandibular disorders DC/TMD, and the Fonseca anamnestic index [20,48].

There are many conditions that simulate TMDs, like clenching, bruxism, trauma or luxation, maxillary sinusitis, or neuralgia [49]. To date, however, TMD is related to the patient-reported symptoms and the objective clinical gnathological consultation. Patients often referred reduced mandibular movements, masseter or temporal hypertrophy, and preauricular pain [50]. If the orofacial pain is not influenced by mandibular excursion, a different etiology might be suspected, while cases of clicking could also occur in asymptomatic patients [51]. Malocclusions, parafunctions, or edentulism should also be considered [52].

TMJ presents unique anatomical and biomechanical characteristics that make it susceptible to a variety of pathological conditions, including osteoarthritis (OA), chronic inflammatory arthritis (CIA), and other degenerative or inflammatory disorders [53]. Although not a typical weight-bearing joint, TMJ is subject to significant shear forces due to its complex motion, which can lead to progressive structural changes over time. The involvement of TMJ in conditions such as rheumatoid arthritis (RA), axial spondyloarthritis (r-ax SpA), and psoriatic arthritis (PsA) is well-documented, with histopathological alterations including synovial proliferation, fibrocartilage changes, erosions, and subchondral bone sclerosis [54].

These pathological changes often result in functional alterations which can be reflected in the bioelectrical activity of the associated masticatory muscles. Notably, synovitis and joint degeneration have been linked to modifications in EMG signals, particularly in muscles such as the lateral pterygoid. Since TMJ dysfunction affects both sides of the joint complex due to its mechanical interdependence, even unilateral pathology may lead to compensatory changes in the contralateral musculature, further influencing EMG patterns [55]. Understanding these bioelectrical alterations is crucial for refining the clinical utility of EMG in the assessment of TMJ disorders, reinforcing the need for a multidisciplinary approach that considers both joint and muscular involvement in disease progression.

The initial diagnosis should involve radiographs (transcranial and trans-maxillary views) or a panoramic view to provide additional information, while magnetic resonance imaging (MRI) is the best way to evaluate joints state in subjects with signs of TMD [31]. However, false-positive results are reported in 20–34% of all asymptomatic patients, and MRI is reserved to patients with ongoing problems in cases of useless conservative treatment [56].

sEMG is the evaluation of electrical signals produced by muscle activity during contraction in the performance of dynamic and static tasks [48]. According to Klasser and Okeson [57,58], the importance of sEMG in detecting and managing TMD is very limited, and could even lead to avoidable dental treatment. Several studies have been conducted to understand whether the application of sEMG can objectively estimate TMD presence [31,36,44,45,59,60,61].

The clinical utility of sEMG as a diagnostic tool also varies across medical specialties, depending on the specific pathology being evaluated. sEMG has been widely investigated and applied in dentistry and maxillofacial surgery for assessing TMDs [58], but its relevance in other specialties, such as rheumatology, may be more limited. Rheumatologists primarily encounter TMJ involvement in the context of CIA or OA, where morphological imaging and other diagnostic criteria are often prioritized [62]. Consequently, with these pathologies, sEMG may not be a routine tool.

However, for specialists managing primary TMJ disorders, including disc displacement and muscle dysfunction, sEMG may provide additional diagnostic insights. This highlights the importance of considering the applicability of diagnostic methods within the scope of different specialties, as the perceived utility of sEMG is inherently dependent on the clinical context and the type of patients encountered.

In a recent systematic review, it was evidenced that patients with diagnosed TMD with pain may change masticatory muscle activation due to sensorimotor interactions and also affect recordings. The symptom of pain could lead to excessive variability and compromise the accuracy of evaluations, and EMG screening of subjects with painful TMD therefore has limitations. The clinical efficiency of sEMG in the detection of this pathological status has not yet been well researched [63]. Furthermore, the fact that there are no reference standards on which the differential diagnosis between TMD and asymptomatic subjects can be based is a limitation to the exclusivity application in clinical practice [59].

Another recent systematic review has highlighted the high heterogeneity of findings related to muscle pattern activations in TMD patients by conducting a meta-analysis on some of the most common sEMG parameters [48]. As in our review, the authors included only studies that had diagnosed TMD in agreement with the DC/TMD criteria to reduce the risk of bias and studies with a control population with healthy TMJ to compare the muscle signals. As with our evaluations, common limitations were found in relation to the high variability present in the studies: the type of electromyographic instrument and test (static or dynamic) used, and the characteristics of the populations included.

Our meta-analyses were carried out in the presence of perfect homogeneity of the tests analyzed and parameters evaluated: static indexes (IMP, POC, and torque coefficient), and dynamic index SMI. According to our analysis, patients with TMD had lower values than healthy ones in all parameters except IMP in relation to the increased probability of developing parafunctions [45,64], but only two studies were included in the meta-analysis [44,45] and the heterogeneity was high with no statistically significant effect size. On the other hand, according to the pain adaptation model, muscle activity is reduced to preserve the sensorimotor system from damage to muscle tissue [26,65,66]. The fact that the IMP does not decrease, or even increases, in dysfunctional subjects confirms the diagnosis of TMD. Recruitment inhibition caused by the avoidance reflex does not occur in TMD patients, who contract significantly and have no protective inhibition; thus, overloads of symptomatic patients, even in the presence of occlusal instability, result in high IMP values [67,68]. Some authors reported that using an active disturbance caused a marked decrease in the frequency and intensity of contraction episodes in healthy patients, but not in subjects with diagnosed TMD or parafunction [69].

SMI value was statistically lower than the controls, in agreement with other findings in the literature [26,70]. This is probably due to the reduced masticatory efficiency of TMD patients compared to healthy ones. The POC index (unit, %) instead assesses the symmetric distribution of muscular activity as determined by occlusion [31]. In our meta-analysis, TMD patients showed lower values than healthy ones, and there was homogeneity between the studies and the effect size was statistically significant, which means that TMD results in a reduction in symmetry in muscle activity, which may be due to functional impairment or compensatory muscle mechanisms as a result of pain. In particular, lower SMI and POC values can be explained by several factors. Firstly, TMD patients exhibit altered activation patterns of the masticatory muscles, with modified muscle recruitment, particularly between the temporalis and masseter muscles, compared to healthy individuals. Additionally, the masticatory muscles in TMD patients tend to be more hypertonic at rest, less efficient, and more prone to fatigue than those of healthy controls. These alterations in muscle function and coordination can result in increased asymmetry in muscle contraction between the right and left sides, leading to lower SMI and POC values, which reflect reduced symmetry and compromised efficiency in muscle activity.

According to other studies [35,45], no significant discrepancy in POC was detected in TMD patients and controls, and it is relevant to mention that our findings may be affected by the heterogeneity of the TMD patients included in the POC meta-analysis (osteoarthritis, soft tissue damage, moderate or severe TMD). It should also be noted that for certain indices such as IMP, only a relatively small number of studies were included. This limitation may affect the precision and reliability of the conclusions drawn, indicating the need for further research to confirm these findings.

However, the available data did not provide sufficient clarity to fully support the primary objective of this study related to differentiations between articular and muscular TMD types using sEMG due to the heterogeneity of the included studies, as well as the limited number of studies specifically addressing these subgroups. The articles are difficult to compare due to the excessively heterogeneous populations (children, students, women, or men) and type of electromiographical instrumentation used, as reported in Table 2; moreover, many authors do not distinguish between pathological subjects with joint, muscle, or mixed-nature TMDs. Of course, methodological factors associated with sEMG recordings, like impedance imbalance susceptivity, may also compromise the accuracy of this method and be the reason for the conflicting findings presented in the literature [22].

The clinical application of diagnostic methodologies, such as EMG, often follows an evolutionary trajectory, moving through phases of initial validation, increasing adoption, critical reassessment, and eventual refinement of their role in clinical practice. This pattern is also evident in the literature referenced in this review, which reflects both early enthusiasm and later evaluations of strengths and limitations. In the case of sEMG, its use has expanded across various medical fields, with an increasing number of studies exploring its potential applications. However, as with many methodologies, the identification of limitations has led to a more refined understanding of its role, guiding its appropriate application in specific clinical contexts. This cyclical evolution highlights the importance of continuously reassessing diagnostic tools to further clarify the specific conditions in which sEMG provides the greatest clinical benefit, optimizing its integration into multidisciplinary diagnostic frameworks.

Nevertheless, our findings suggest that sEMG may still offer valuable insights into the overall diagnosis of TMD, warranting further investigation in future studies with more specific patient populations and refined diagnostic criteria.

## 5. Conclusions

sEMG has been widely used to evaluate the electrophysiological behaviour of muscles as a non-invasive instrumentation that is also easy to use and accessible, but the discriminatory ability on joint, muscle, or mixed-origin TMDs need to be considered for future studies. EMG is not able to distinguish between healthy or unhealthy patients, but there might be better results if employed in muscle forms [59]. Subjects with TMD, myogenous TMD, with DD without pain, and also children with TMD, showed a lower EMG activity than controls. Patients with TMD had lower values in the dynamic index SMI and in the static indexes POC and torque coefficient, except IMP.

Although this study has limitations, particularly in terms of the need for a more diverse patient sample categorized according to specific etiology and international classification criteria, the current meta-analysis demonstrates that sEMG, when assessing the POC and SMI parameters, remains a reliable tool. This supports the potential of sEMG as a useful diagnostic method, encouraging further in-depth studies aimed at enhancing its ability to distinguish between healthy and pathological patients. Such research is essential to optimize the predictive value of EMG in clinical practice.

## Figures and Tables

**Figure 1 healthcare-13-00466-f001:**
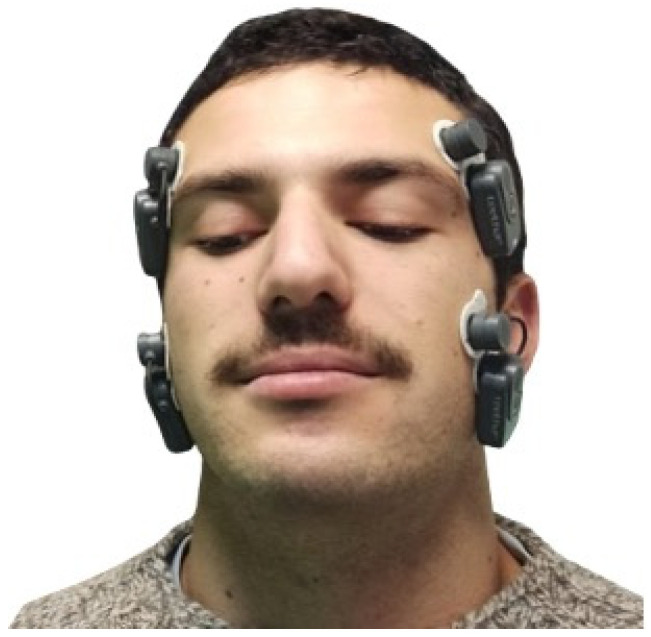
sEMG electrode placement.

**Figure 2 healthcare-13-00466-f002:**
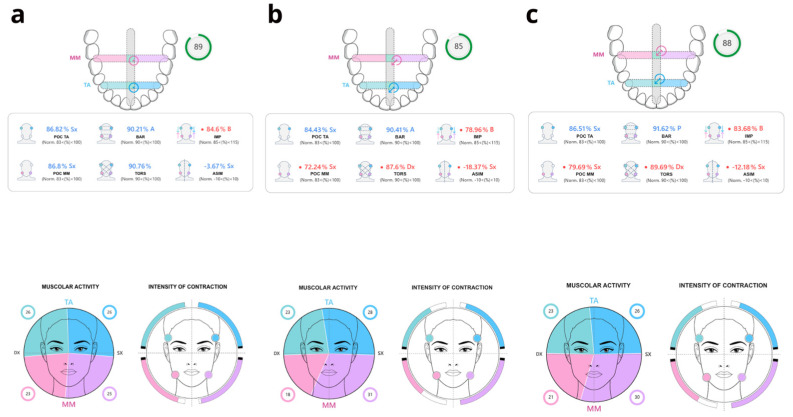
Clinical report on the indexes values and occlusal balance of a subject with articular TMD, related to the following subjects: (**a**) healthy; (**b**) with muscular TMD; (**c**) with articular TMD. Masseter muscles (MM); anterior temporalis (TA); percent Overlap Coefficient (POC); barycenter (BAR); torsion (TORS); musclework (IMP); asymmetry (ASIM); Normality (Norm.) [23].

**Figure 3 healthcare-13-00466-f003:**
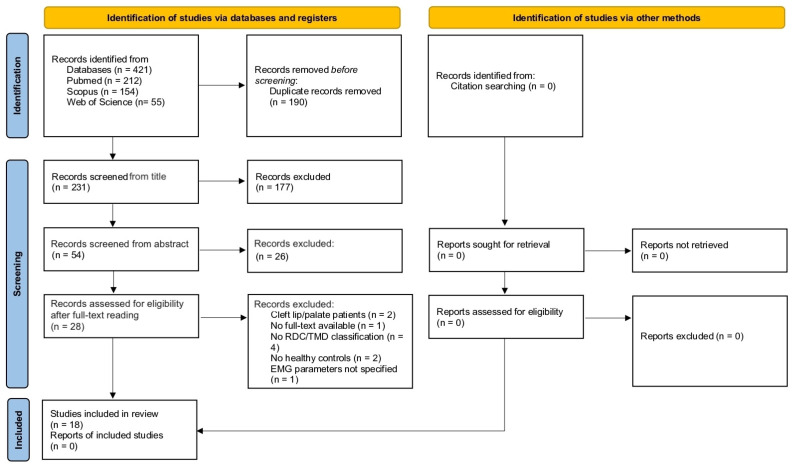
Flow diagram of the included studies.

**Figure 4 healthcare-13-00466-f004:**
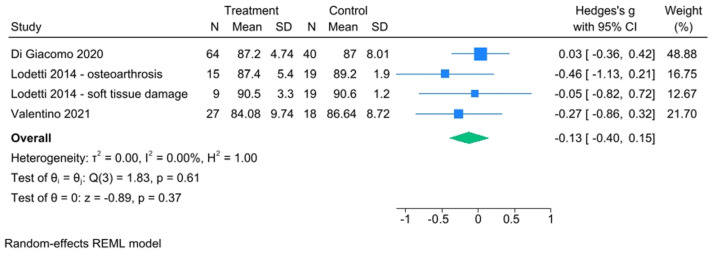
Torque forest plot [31,44,45].

**Figure 5 healthcare-13-00466-f005:**
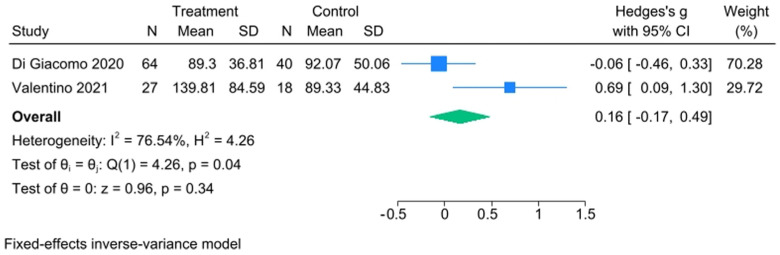
IMP forest plot [44,45].

**Figure 6 healthcare-13-00466-f006:**
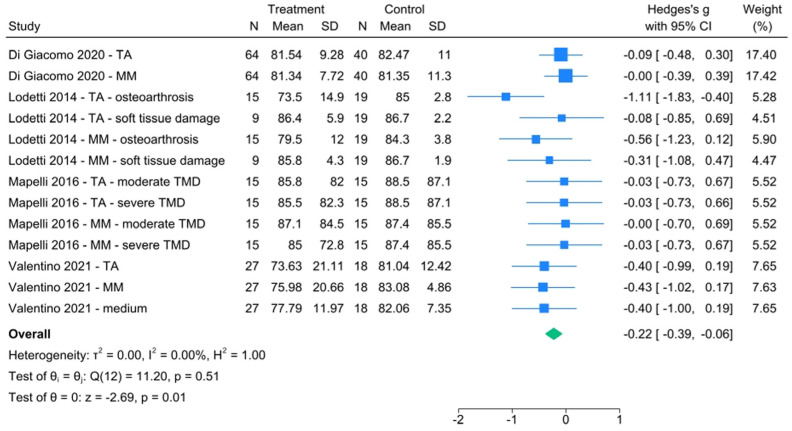
POC forest plot [31,35,44,45].

**Figure 7 healthcare-13-00466-f007:**
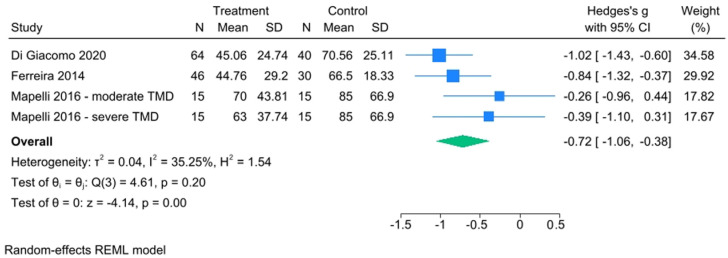
SMI forest plot [30,35,44].

## Data Availability

The original contributions presented in this study are included in the article/Appendix A. Further inquiries can be directed to the corresponding author.

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
