# Peer review of "Systematic Review and Meta-Analysis of Electromyography Potential to Discriminate Muscular or Articular Temporomandibular Disorders and Healthy Patients"

_healthcare, 2025, doi:10.3390/healthcare13050466_

Round 1
Reviewer 1 Report
Comments and Suggestions for Authors
This systematic review makes a valuable contribution to understanding the diagnostic potential of sEMG for TMDs. I offer the following suggestions for improvement:
1. The paper seems quite arid. To make the paper more intuitive and visually appealing, the authors could include visuals such as an annotated diagram of sEMG electrode placement to illustrate the methodology, or a comparative bar chart or heatmap showing key findings (e.g., POC, SMI, and IMP differences between TMD and healthy patients). Or a diagram showing muscle activity patterns in healthy vs. TMD patients.
2. The authors state that the initial diagnosis should involve radiographs (transcranial and trans-maxillary views) or a panoramic view to provide additional information, followed by MRI, which has high costs and, as noted, a 20%-34% false-positive rate in asymptomatic patients, making it reserved for patients with ongoing issues after conservative treatment. However, the paper does not address the role of ultrasound, a frequently employed diagnostic tool in TMDs. I suggest the authors consider discussing the utility of ultrasound in TMD diagnosis, as it offers a non-invasive, cost-effective alternative that has shown value in evaluating soft tissue conditions.
Please see this paper: Popescu MN, Beiu C, Iliescu CA, Racoviță A, Berteanu M, Iliescu MG, Stănescu AMA, Radaschin DS, Popa LG. Ultrasound-Guided Botulinum Toxin-A Injections into the Masseter Muscle for Both Medical and Aesthetic Purposes. Toxins (Basel). 2024 Sep 24;16(10):413. doi: 10.3390/toxins16100413. PMID: 39453189; PMCID: PMC11511025.
Author Response
comment 1: The paper seems quite arid. To make the paper more intuitive and visually appealing, the authors could include visuals such as an annotated diagram of sEMG electrode placement to illustrate the methodology, or a comparative bar chart or heatmap showing key findings (e.g., POC, SMI, and IMP differences between TMD and healthy patients). Or a diagram showing muscle activity patterns in healthy vs. TMD patients.
Response 1: Thank you for this valuable suggestion. We have now included a figure of sEMG electrode placement (new Figure 1), as well as a figure (new Figure 2) related to examples of EMG report with parameters recordings and muscle activity patterns of different subjects.
comment 2: The authors state that the initial diagnosis should involve radiographs (transcranial and trans-maxillary views) or a panoramic view to provide additional information, followed by MRI, which has high costs and, as noted, a 20%-34% false-positive rate in asymptomatic patients, making it reserved for patients with ongoing issues after conservative treatment. However, the paper does not address the role of ultrasound, a frequently employed diagnostic tool in TMDs. I suggest the authors consider discussing the utility of ultrasound in TMD diagnosis, as it offers a non-invasive, cost-effective alternative that has shown value in evaluating soft tissue conditions.
Please see this paper: Popescu MN, Beiu C, Iliescu CA, Racoviță A, Berteanu M, Iliescu MG, Stănescu AMA, Radaschin DS, Popa LG. Ultrasound-Guided Botulinum Toxin-A Injections into the Masseter Muscle for Both Medical and Aesthetic Purposes. Toxins (Basel). 2024 Sep 24;16(10):413. doi: 10.3390/toxins16100413. PMID: 39453189; PMCID: PMC11511025.
Response 2: We greatly appreciate this recommendation and have now incorporated a paragraph on ultrasound as a diagnostic tool for TMD in the Introduction section. We have cited the suggested paper (Popescu et al., 2024) to support this information, as follows:
“In particular, ultrasound, a non-invasive and cost-effective tool, has a clinically acceptable ability to diagnose disc displacement and joint effusions in patients with TMD [12] and is also used to guide the operator in botulinum masseter injections with the aim of reducing muscular TMD symptoms, managing oromandibular dystonia, myofascial pain syndrome and tension-type head-aches. Ultrasound allows visualization of the precise injection site, ensuring accurate delivery of botulinum to masseter muscle, avoiding surrounding structures and mini-mizing adverse effects and is also a quantitative measurement tool, able to compare the vertical height of the masseter muscle before and after botulinum injection [13].”
Reviewer 2 Report
Comments and Suggestions for Authors
A few issues exist
(The Authors must see my remarks)

Author Response
Comment 1: Add reference
Comment 2: Add reference
Response 1 and 2: We greatly appreciate this suggestions, and relevant references have been inserted in the suggested section, as follows:
Line 141-147: Page, M.J.; McKenzie, J.E.;, Bossuyt, P.M.; et al. The PRISMA 2020 statement: an updated guideline for reporting systematic reviews. BMJ. 2021;372:n71.
Line 137-140: Methley, A.M.; Campbell, S.; Chew-Graham, C.; McNally, R.; Cheraghi-Sohi, S. PICO, PICOS and SPIDER: A Comparison Study of Specificity and Sensitivity in Three Search Tools for Qualitative Systematic Reviews. BMC Health Serv. Res. 2014;14:579
Reviewer 3 Report
Comments and Suggestions for Authors
WORD file

Author Response
Comment 1: The importance of each diagnostic method must be evaluated through the prism of the different pathology that is the subject of different medical specialties, and accordingly the method has a specific value in the work of different medical specialists. It is pertinent to note here that this reviewer is a rheumatologist and despite his 30 years of clinical experience, he has needed intramuscular EMG as a differential diagnostic tool in the field of TMJ pathology only 5 times (on average once every 5 years) when the patient's complaints did not correspond to the degree of morphological changes in the joint and a further reason was sought for it. This is, of course, because rheumatologists encounter regional TMJ problems -mainly in the course of chronic inflammatory arthritis (CIA) or as an additional localization in the course of osteoarthritis (OA), and patients with isolated TMJ complaints - quite reasonably fall to maxillofacial surgery specialists or dentists. That is, each methodology and specifically sEMG has different usefulness for different specialists encountering the same regional problem, but with different etiology.
Response 1:
We sincerely appreciate the reviewer's insightful comments regarding the varying importance of different diagnostic methods across medical specialties. We acknowledge that the clinical utility of sEMG differs depending on the specific pathology and the specialist involved. Our study follows diagnostic recommendations which are primarily aligned with the perspectives of dental and craniofacial specialists.
We recognize that, from a rheumatological standpoint, sEMG may have limited application in TMJ pathology, particularly within the context of chronic inflammatory arthritis and osteoarthritis. However, our study is primarily intended to evaluate its relevance within the field of dentistry and maxillofacial surgery, where its diagnostic value may be more pronounced.
we have now added new paragraphs in the introduction section that concerns the role of electromyography as a diagnostic tool for different pathologies pertaining to specific medical specialties as follows:
“EMG has been widely utilized across various medical fields, including neurology, psychology, psychiatry, physiatry, physical medicine, rehabilitation, kinesiology, and gnatology. Each discipline leverages specific aspects of this technique based on its clinical needs and the degree of clinical acceptance of EMG varies depending on its application.
In kinesiology, sEMG is a well-established tool for analyzing movement disorders and is particularly useful in differentiating various types of tremors, myoclonus, and dystonia, as well as in evaluating gait and posture. In neurology and rehabilitation, sEMG is most commonly employed in the treatment of weak or paretic muscles due to peripheral nerve injuries [14]. EMG has been widely used in various medical filed is also used to identify different pathologies, like amyotrophic lateral sclerosis, myasthenia gravis due to degeneration of muscle tissue and nerve, for the assessment and neurorehabilitation of cerebral palsy in children [15]. It is used as part of physical therapy to enhance muscle activity and strength, both in post-operative rehabilitation following surgical nerve repair and in non-surgical cases [14], to evaluate muscle fatigue measurements in pre- and post-surgery moni-toring [16], or in orthopedical applications, surgical procedures, nervous system studies, and postural assessment [17].”
And
“While, in rheumatology, electromyography (EMG) is a valuable tool for supporting the diagnosis of inflammatory myopathy or neuropathic processes. It can help differen-tiate active inflammatory myopathy from glucocorticoid-induced myopathy, as virtually all patients with active myositis present with abnormal EMG findings. However, while EMG can confirm the presence of myopathy, it does not provide information on its eti-ology [18]. Regarding EMG findings in patients with osteo-arthritis and rheumatoid arthritis, specific parameters, such as maximum molar bite force and maximum mouth opening range, may differ from those observed in healthy indi-viduals [19]”
Comment 2: Naturally, the authors are not to blame for this, but only the editors who made the suggestion and the reviewer himself, who accepted the commitment, mainly for the purpose of comparing his own clock (visions) with that of the dentists or craniofacial surgery specialists. We will continue with the fact that the manuscript submitted for review was prepared in unison with the diagnostic recommendations of International Association for Dental Research and International Association for the Study of Pain, specified in the author's reference [6]. But these same recommendations are just a curiosity (another look at a regional problem) in the eyes of a rheumatologist considering TMJ-pathology in the context of mentioned above CIA, OA, primary and metastatic tumors or rare trauma (in ours pracice) and TMJ -dysplasias, disc displacement etc.
Response 2:
We have clarified in the discussion section that the applicability of sEMG is specialty-dependent and may not be equally relevant for all medical professionals managing TMJ disorders. We are grateful for this perspective, which helps us refine the scope and interpretation of our findings, as follows:
“The clinical utility of sEMG as a diagnostic tool also varies across medical specialties, depending on the specific pathology being evaluated. sEMG has been widely investi-gated and applied in dentistry and maxillofacial surgery for assessing TMDs [58], but its relevance in other specialties, such as rheumatology, may be more limited. Rheumatologists primarily encounter TMJ in-volvement in the context of chronic inflammatory arthritis (CIA) or osteoarthritis (OA), where morphological imaging and other diagnostic criteria are often prioritized [62]. Consequently, with these pathologies sEMG may not be a routine tool.
However, for specialists managing primary TMJ disorders, including disc displacement and muscle dysfunction, sEMG may provide additional diagnostic insights. This highlights the importance of considering the applicability of diagnostic methods within the scope of different specialties, as the perceived utility of sEMG is inherently dependent on the clinical context and the type of patients encountered.”
Comment 3: The meaning of each methodology over the time is also different and most generally follows the course of the pendulum. It begins with validation of the methodology and increases in use and accumulation of experience and information (increasing number of publications); revealing weaknesses (withdrawing from the methodology, reducing publications); specifying the exact place and role of the methodology (new increase in publications). This is also evident from the referenced publications used by the authors in this systematic review and its meta-analyses.
Response 3:
We appreciate the reviewer’s insightful perspective on the evolving significance of different methodologies over time. We acknowledge that scientific methodologies often follow a cyclical pattern of validation, widespread adoption, critical reassessment, and eventual refinement of their role within the field. In our systematic review and meta-analysis, we aimed to capture the most recent and relevant evidence regarding the use of sEMG in specific clinical application, recognizing that its clinical utility may continue to evolve as more research emerges.
We also agree that the referenced publications reflect this progression, as they include studies from different phases of methodological development. To address this, we will briefly mention in the discussion that the role of methodology has evolved over time and continues to be refined based on accumulating evidence and technological advancements. Thank you for this valuable observation, which helps us further contextualize our findings.
We have added a paragraph at the end of the discussion section as follows:
“The clinical application of diagnostic methodologies, such as EMG, often follows an evolutionary trajectory, moving through phases of initial validation, increasing adoption, critical reassessment, and eventual refinement of their role in clinical practice. This pattern is also evident in the literature referenced in this review, which reflects both early enthusiasm and later evaluations of strengths and limitations. In the case of sEMG, its use has expanded across various medical fields, with an increasing number of studies exploring its potential applications. However, as with many methodologies, the identification of limitations has led to a more refined understanding of its role, guiding its appropriate application in specific clinical contexts. This cyclical evolution highlights the importance of continuously reassessing diagnostic tools to further clarify the specific conditions in which sEMG provides the greatest clinical benefit, optimizing its integration into multidisci-plinary diagnostic frameworks.”
Comment 4: When we talk about TMJ, some anatomical-physiological features (APF) are important in connection with the pathology affecting the joit. The TMJ is a complex synovial joint, The typical loading for the joint is shear stress occurring during the twisting and sliding motions to which the fibrocartilaginous disc and articular condyle are subjected. Although the TMJ is not a typical "weight-bearing joint" like the knee or hip joints, the TMJ is notorious for its vulnerability to wear and tear (OA of the TMJ), which is an almost universal finding as we age. Additional precipitating factors here are the various deviations in the joint surfaces and intra-articular disc (disc displacements and damages, trauma, avascular necrosis, TMJ-dysplasias) and calcium monophosphate and pyrophosphate deposits, giving flares in the course of OA of the joint. The joint is also typically involved in the course of CIA, being targeted in RA, r-ax SpA (former AS), PsA and sometimes in reactive arthritis. The most common histopathological changes are synovial proliferation and fibrocartilage proliferation of the mandibular head and articular eminences. Erosions (RA, r-ax SpA, PsA), joint space narrowing, subchondral bone sclerosis and cysts in the later stages can also be found. In addition, the joints can be affected by infections, avascular necrosis, primary and metastatic tumors and trauma. All mentioned etiological noxes ultimately lead to internal derangement with impaired function and a differently expressed local inflammatory process. A key point for any pathological process affecting the TMS is that the joints on both sides of the body are mechanically connected, i.e. limitation of motion in one joint (regardless of the cause) inevitably affects the other, even though the joint on the other side may be normal. This is accompanied by changes in the fibrocartilage, which forms the articular surfaces of the skull and mandible, and is also represented rich in by an intra-articular disc that divides the joint cavity into upper and lower parts (reason for spondyloarthropathies /SpA/ to perceive the TMJ as a target). bioelectrical activity of the muscles moving the joint, i.e. -altered bioelectrical activity can be recorded from the healthy joint. The second key point is that any local inflammatory process (synovitis), depending on its intensity, is associated with a change in the bioelectrical characteristics of the regional muscle groups (mainly the lateral pterygoid muscle), i.e. -inflammatory processes of different etiology (RA, PsA, crystal induced flares in OA, esptic arthritis etc.) can be accompanied by similar bioelectrical changes from the mentioned muscles.
Response 4:
We appreciate the reviewer’s detailed insights into the anatomical and pathophysiological features of the TMJ and their relevance to various pathological conditions. We fully acknowledge that the TMJ, despite not being a typical weight-bearing joint, is susceptible to degenerative changes and inflammatory involvement, particularly in the context of osteoarthritis (OA) and chronic inflammatory arthritis (CIA), including rheumatoid arthritis (RA), axial spondyloarthritis (r-ax SpA), and psoriatic arthritis (PsA).
The connection between TMJ dysfunction and altered muscle bioelectrical activity is particularly relevant. As the reviewer pointed out, synovitis and structural joint alterations can influence the bioelectrical properties of regional muscle groups, particularly the lateral pterygoid muscle, potentially leading to measurable changes in EMG signals. This observation aligns with the rationale behind our analysis, which aims to assess the utility of EMG in evaluating TMJ dysfunction across different pathological conditions.
We will ensure that the discussion appropriately reflects these key anatomical and pathophysiological considerations, further clarifying how TMJ pathology—whether inflammatory, degenerative, or mechanical—may impact muscle bioelectrical activity. Thank you for this valuable contribution, which helps enhance the contextualization of our findings.
We added a paragraph in the discussion section as follows:
“TMJ presents unique anatomical and biomechanical characteristics that make it susceptible to a variety of pathological conditions, including osteoarthritis (OA), chronic inflammatory arthritis (CIA), and other degenerative or inflammatory disorders [53]. Although not a typical weight-bearing joint, TMJ is subject to significant shear forces due to its complex motion, which can lead to progressive structural changes over time. The involvement of TMJ in conditions such as rheumatoid arthritis (RA), axial spondyloarthritis (r-ax SpA), and psoriatic arthritis (PsA) is well-documented, with histopathological alterations including synovial proliferation, fibrocartilage changes, erosions, and subchondral bone sclerosis [54].
These pathological changes often result in functional alterations, which can be reflected in the bioelectrical activity of the associated masticatory muscles. Notably, synovitis and joint degeneration have been linked to modifications in EMG signals, particularly in muscles such as the lateral pterygoid. Since TMJ dysfunction affects both sides of the joint complex due to its mechanical interdependence, even unilateral pathology may lead to compensatory changes in the contralateral musculature, further influencing EMG patterns [55]. Understanding these bioelectrical alterations is crucial for refining the clinical utility of EMG in the assessment of TMJ disorders, reinforcing the need for a multidisciplinary approach that considers both joint and muscular involvement in disease progression.”
All this was mentioned by the reviewer because one methodology has different value both for different specialties and over time (X-Ray of TMJ vs. MRI). The strengths of the methodology (detection of disturbances in the transmission of impulses and/or their transformation in the neuromuscular synapse and the quality of the muscle response) must be known and used correctly because artificially expanding the indications for the methodology turns them into weaknesses.
That is, the reviewer has no remarks regarding the choice of methodology (systematic review with meta- analyses) and the results obtained from the two processes. But he has remarks on the introduction part (where everything mentioned above about the value of the methodology should be commented on) and on the discussion part (which should also be done through the prism of the comments mentioned for the introduction), further pointing out the weaknesses mentioned of the methodology.
We sincerely appreciate the reviewer time and effort in evaluating our manuscript, and believe that the revisions have significantly strengthened our manuscript. The feedback has significantly helped us refine our work. We have carefully addressed all comments and made the necessary revisions in the point-by-point response.
Reviewer 4 Report
Comments and Suggestions for Authors
The manuscript is a systematic review and a meta-analysis assessing the role of surface electromyography sEMG in the diagnosis of healthy subjects versus those with temporomandibular disorders (TMDs). The paper is well developed, following PRISMA guidelines. The clinical problem addressed in the study is pertinent; however, there is a significant lack of clarity, precision, and scientific impact that needs to be improved upon in the paper.
There is a need to articulate the objectives more clearly as the aim has been stated. There are gaps in the methodology and the results. For example, the objective of the study is to differentiate between joint and muscles TMD types. However, the objectives are not met in the results and discussion.
While there are criteria for including studies, there needs to be justification provided on the selection of only studies from the last 10 years. Older foundational studies may add useful information.
Though there are mentions of the QuADS tool, there is a substantial lack of explanation in relation to the actual scoring, which affects how the study was selected or not selected.
Discrepancies related to data extraction are not elaborated on sufficiently. Increased clarity with this process is necessary
The analysis of variation has been conducted with the help of Hedge’s g and the use of random/fixed effects models are applied properly. However:
Some indices (e.g., IMP) have a very small number of studies which may affect the precision of the conclusions that can be drawn. Explicitly put that this might be a limitation.
Funnel plots were only mentioned without delving into them, as other tests of bias such as Egger’s test may be helpful to the analysis.
Trying to fill the gap between the findings and clinical practice is extremely shallow in this section. For instance:
How do clinicians explain the lower levels of SMI and POC in TMD patients?
The statement in the conclusion, where it is asserted that sEMG is ‘not reliable’ but ‘encouraging’ for some indices, is quite vague on the surface and needs to be refined at the least.
There is limited comparison with previous reviews. It would be better if gaps and similarities with other systematic reviews were addressed.
Heterogeneity between studies is recognized, but it would need more examination. The authors should provide more details on the reasons for the differences (e.g., devices, populations, and TMDs).
The Figure 1 PRISMA Flow Diagram is very efficient, but fails to give the rationale behind the exclusion of studies at these stages.
Figure 2, Figure 3, Figure 4 and Figure 5 are now inserted as images. It should be written as tables with text.
Author Response
The manuscript is a systematic review and a meta-analysis assessing the role of surface electromyography sEMG in the diagnosis of healthy subjects versus those with temporomandibular disorders (TMDs). The paper is well developed, following PRISMA guidelines. The clinical problem addressed in the study is pertinent; however, there is a significant lack of clarity, precision, and scientific impact that needs to be improved upon in the paper.
Comment 1: There is a need to articulate the objectives more clearly as the aim has been stated. There are gaps in the methodology and the results. For example, the objective of the study is to differentiate between joint and muscles TMD types. However, the objectives are not met in the results and discussion.
Response 1: We thank the reviewer for the comment, we have revised the discussion section as suggested to enhance clarity and precision, adding information as follows:
“However, the available data did not provide sufficient clarity to fully support the primary objective of this study related to differentiate between articular and muscular TMD types using sEMG, due to the heterogeneity of the included studies, as well as the limited number of studies specifically addressing these subgroups.
[…]
Nevertheless, our findings suggest that sEMG may still offer valuable insights into the overall diagnosis of TMD, warranting further investigation in future studies with more specific patient populations and refined diagnostic criteria.”
Comment 2: While there are criteria for including studies, there needs to be justification provided on the selection of only studies from the last 10 years. Older foundational studies may add useful information.
Response 2: Thank you for your comment. We selected studies from the last 10 years to ensure that the evidence analyzed is up-to-date and reflects the most recent knowledge in the field. We acknowledge the value of foundational studies published earlier, anyway given the rapid evolution of research in this area, older studies may not fully capture recent methodological developments, emerging trends, or new theoretical perspectives. For this reason, we have focused on more recent literature to maintain the relevance and applicability of our analysis.
Comment 3: Though there are mentions of the QuADS tool, there is a substantial lack of explanation in relation to the actual scoring, which affects how the study was selected or not selected.
Response 3: We have included a detailed explanation at the end of materials and methods section of the QuADS tool, describing how scoring was applied in study selection, and in Figure S5 are reported the complete characteristics of the tool and the scoring related to the 13 criteria.
Comment 4: Discrepancies related to data extraction are not elaborated on sufficiently. Increased clarity with this process is necessary.
Response 4: We have clarified the process of data extraction and how discrepancies were resolved, as follows:
“Articles that met the inclusion criteria were first screened based on their abstracts. The full texts of potentially eligible articles were then reviewed independently by four reviewers. Data extraction was also performed independently by the same reviewers, and any discrepancies or uncertainties were resolved through discussion with the other authors to reach a consensus. Final decisions were made through group discussions, ensuring consistency and accuracy in the extracted data. Following the electronic search, the reviewers conducted a manual search of the reference lists of the included articles, ap-plying the same inclusion/exclusion criteria.
Relevant data were organized into two tables, including:
Table 1: Author(s), year of publication, EMG parameters evaluated, and types of EMG tests analyzed; and Table 2: Country, number of cases and controls, type of electromyography device, diagnostic criteria, inclusion and exclusion criteria, main results, conclusions, and funding sources.”
Comment 5: The analysis of variation has been conducted with the help of Hedge’s g and the use of random/fixed effects models are applied properly. However:
Some indices (e.g., IMP) have a very small number of studies which may affect the precision of the conclusions that can be drawn. Explicitly put that this might be a limitation.
Response 5: Thank you for your comment. We acknowledge that the small number of studies for IMP index may affect the precision of our conclusions. To address this, we have explicitly stated this as a limitation at the end of the discussion section and have emphasized the need for further research to strengthen the evidence base, as follows:
“It should be noted that for certain indices, such as IMP, only a relatively small number of studies were included. This limitation may affect the precision and reliability of the conclusions drawn, indicating the need for further research to confirm these findings.”
We appreciate your suggestion, which has helped improve the clarity of our manuscript.
Comment 6: Funnel plots were only mentioned without delving into them, as other tests of bias such as Egger’s test may be helpful to the analysis.
Response 6: Thank you for considering this issue. We acknowledge that a deeper exploration of funnel plots and additional tests for bias, such as Egger’s test, could provide further insights into the analysis. However, given the nature of the data and the scope of our study, we opted not to include Egger’s test. We have clarified this in the revised manuscript at the end of the Results section, as follows:
“Regarding the funnel plots (Figures S1-S4), the same analysis was applied to all study groups, and none of the plots suggest significant publication bias. The only exception is the POC analysis (Figure S3), where one study falls outside the confidence interval. However, this was not considered a substantial issue, and therefore, additional analyses such as Egger's test were not performed”
Comment 7: Trying to fill the gap between the findings and clinical practice is extremely shallow in this section. For instance:
How do clinicians explain the lower levels of SMI and POC in TMD patients?
Response 7: We have expanded the Discussion to better connect findings with clinical implications, explaining how lower SMI and POC levels in TMD patients are interpreted clinically, as follows:
“In particular, lower SMI and POC values can be explained by several factors. Firstly, TMD patients exhibit altered activation patterns of the masticatory muscles, with modified muscle recruitment, particularly between the temporalis and masseter muscles, compared to healthy individuals. Additionally, the masticatory muscles in TMD patients tend to be more hypertonic at rest, less efficient, and more prone to fatigue than those of healthy controls. These alterations in muscle function and coordination can result in increased asymmetry in muscle contraction between the right and left sides, leading to lower SMI and POC values, which reflect reduced symmetry and compromised efficiency in muscle activity.”
Comment 8: The statement in the conclusion, where it is asserted that sEMG is ‘not reliable’ but ‘encouraging’ for some indices, is quite vague on the surface and needs to be refined at the least.
Response 8: We have revised the Conclusion to clarify the statement on sEMG reliability, emphasizing its potential while acknowledging its limitations, as follows:
“Although the study has limitations, particularly in terms of the need for a more diverse patient sample categorized according to specific etiology and international classification criteria, the current meta-analysis demonstrates that sEMG, when assessing the POC and SMI parameters, remains a reliable tool. This supports the potential of sEMG as a useful diagnostic method, encouraging further in-depth studies aimed at enhancing its ability to distinguish between healthy and pathological patients. Such research is essential to optimize the predictive value of EMG in clinical practice.”
Comment 9: There is limited comparison with previous reviews. It would be better if gaps and similarities with other systematic reviews were addressed.
Response 9: Thank you to have pointed out this problem. We appreciate your suggestion to address the gaps and similarities with other systematic reviews. However, after an extensive search of the literature, we were unable to identify other systematic reviews specifically focused on the differential diagnosis of healthy and pathological subjects using sEMG. As a result, we were not able not include comparisons in our manuscript, with the exception of articles cited in the discussions as a comparison to articles included in the systematic review. We believe this highlights a gap in the current literature, which our study aims to address. We hope this explanation clarifies our approach, and we are open to any further suggestions.
Comment 10: Heterogeneity between studies is recognized, but it would need more examination. The authors should provide more details on the reasons for the differences (e.g., devices, populations, and TMDs).
Response 10: Thank you for your constructive comment. We acknowledge that the heterogeneity between studies is a limitation of our analysis. While we have mentioned this aspect, we agree that a more detailed exploration of the reasons for these differences (e.g., devices, populations, and types of TMDs) would provide a clearer understanding of the variability observed. Due to the diversity in study designs and the heterogeneity of the included populations, it was challenging to provide a comprehensive examination of all the contributing factors. However, we have now modified Table 2 and limitations section to include these informations. As follows:
“The articles are difficult to compare, due to the excessively heterogeneous populations (children, students, women, or men) and type of electromiographical instrumentation used, as reported in Table 2; moreover, many authors do not distinguish between pathological subjects with joint, muscle or mixed nature TMDs.”
We hope this clarifies our approach, and we appreciate your suggestion for further refinement.
Comment 11: The Figure 1 PRISMA Flow Diagram is very efficient, but fails to give the rationale behind the exclusion of studies at these stages.
Response 11: We thank the reviewer for the comment, we have corrected the PRISMA Flow Diagram as suggested.
Comment 12: Figure 2, Figure 3, Figure 4 and Figure 5 are now inserted as images. It should be written as tables with text.
Response 12: Thank you for your comment. We understand the suggestion to present Figures as tables with text. However, these figures were generated automatically by the statistical software as forest plots and, due to their format, cannot be easily converted into tables without losing critical information or clarity. Given that these visualizations are an integral part of the statistical analysis, we believe that presenting them as figures is the most appropriate approach. We hope this clarifies our reasoning, and we are open to any further suggestions for improving the presentation.
Round 2
Reviewer 3 Report
Comments and Suggestions for Authors The reviewer thanks the authors for their efforts to improve the quality of the manuscript and has no other significant remarks.Reviewer 4 Report
Comments and Suggestions for Authors
I appreciate the authors for incorporating the review comments.